

# Spring onset and seasonality patterns during the Lateglacial in the eastern Baltic region

Leeli Amon[1], Friederike Wagner-Cremer [2], Jüri Vassiljev[1], Siim Veski[1]

[1]Department of Geology, Tallinn University of Technology, Tallinn, 19086, Estonia
[2]Department of Geosciences and Physical Geography, Utrecht University, Utrecht, 3584, The Netherlands

*Correspondence to*: Leeli Amon (leeli.amon@taltech.ee)

**Abstract.** Spring onset is an important phenological observation that is sensitive to modern climate change and can be traced back in geological time. The Lateglacial (~14500 – 11700 cal yr BP) spring onset and growing season (growth-degree-days)

dynamics in the eastern Baltic region were reconstructed using the micro-phenological approach based on the dwarf birch (*Betula nana*) subfossil leaf cuticles. The presented study sites, Lake Lielais Svetinu (eastern Latvia) and Lake Kosilase (central Estonia), are located ~200 km apart in the region affected by the south-eastern sector of the Scandinavian Ice Sheet. During the Lateglacial period the region and its biota were influenced by the retreating glacier and the different stages of the Baltic Ice Lake. The plant macrofossil data confirms that the study sites were located in different vegetation zones (arctic-to-boreal)

during the Lateglacial period. The dynamics of the estimated length of the growing season and spring onset, combined with the regional collection of plant macrofossil records, suggest the importance of local settings to species migration. During the Lateglacial warming period (Bølling/Allerød), a notable spring warming and longer growing season was calculated based on micro-phenology, but the treeline did not extend beyond central Estonia. The comparison of pollen- and chironomid-inferred past temperature estimations with spring onset, growth-degree-days, and plant macrofossil data shows coherent patterns during

the cooler Older Dryas and warmer Bølling/ Allerød periods while suggesting more complicated climate dynamics and possible warmer episodes during the Younger Dryas cold reversal.

## 1 Introduction

Changing seasonality belongs to the most eminent characteristics of ongoing global climate change. Phenological observations

are sensitive and easily obtainable indicators of biospheric changes in response to climate change (Peñuelas and Filella, 2001; Badeck et al., 2004, Cleland et al. 2007). The earlier unfolding of leaves has been observed since the mid-20th century all over the Northern Hemisphere (Ahas et al., 2002; Badeck et al., 2004 and references therein, Menzel et al. 2006, Jeong et al., 2011 and references therein), although the rate of change has decelerated during the last few decades (Fu et al., 2015). The spring phenology is influenced by pre-season temperature that is described as temperature before spring phenological date, winter



temperature and accumulated precipitation (Wang et al. 2015). The seasonality changes are the subject of intensive discussions, given the strong impact of the earlier spring onset on many sectors including agriculture, ecosystem stability, and vegetation dynamics (e.g. Buermann et al. 2018, Menzel et al. 2020).

Palaeoecological studies have the potential to contribute to a better understanding of spatio-temporal seasonality dynamics by reconstructing seasonal temperature changes during the phases of natural climate change that occurred in the recent past. While

past summer and winter temperatures have been reconstructed more often, the dynamics of the spring season is still underrepresented in the temperature records available so far. The micro-phenological proxy based on the dwarf birch (*Betula nana*) subfossil leaf cuticle features (Wagner-Cremer et al., 2010) offers a means to shed light on past seasonality patterns as well the dynamics of spring onset and the amount of growing-degree-days (GDD). This proxy makes use of the direct correlation between the lateral epidermal cell expansion that regulates cell size and shape during the maturation of leaves

(Wagner-Cremer et al., 2010; Ercan et al., 2020). The longer the growth period available to plants, and the higher the accumulated GDD, the larger and increasingly undulated the epidermal cells grow. Quantified as the Undulation Index (UI) in long-term monitoring studies and validated in free-field growth experiments (Ercan et al., 2021), this proxy has demonstrated asynchronous spring versus summer temperature dynamics during the warming from the Late Pleniglacial to the Bølling/Allerød and during the transition from the Younger Dryas to the Holocene (Wagner-Cremer and Lotter, 2011;

Steinthorsdottir and Wagner-Cremer, 2019). During both warming phases, spring onset and GDD accumulation rose more than a century before the summer temperatures started to increase (Steinthorsdottir and Wagner-Cremer, 2019). A systematic application of this proxy to *B. nana* leaf-bearing lake and peat sequences thus has the potential to add information on the expression of spring season warming over a large geographical range.

In the present study, we focus on the seasonality changes in the eastern Baltic region by compiling multiple temperature and

vegetation reconstructions based on a variety of biological proxies. The Baltic region was covered by the south-eastern sector of the last Scandinavian Ice Sheet (SIS) (Kalm, 2012). During the past decade the Lateglacial environmental history of this region has been intensely studied (e.g. Stancikaite et al., 2009; Heikkilä et al., 2009; Veski et al., 2012; Druzhinina et al., 2020, Seiriene et al., 2020). The Lateglacial changeable climatic conditions (Rasmussen et al., 2014), available past summer and winter temperature reconstructions (Veski et al., 2015), and the associated arctic-to-boreal vegetation dynamics (Amon et al.,

2012; Veski et al., 2012; Amon et al., 2016) and tree-line presence (Amon et al., 2014) provide an excellent case to study the seasonal dynamics of rapid natural climate warming and cooling episodes. Here we apply the microphenological UI proxy to two sites which hold uncommonly continuous and well-preserved *B. nana* subfossil leaf fragments in order to determine spring onset dates and the thermal properties of the growing season. The first indication that the UI proxy is also applicable in a more continental settings has been provided in experimental studies on *B. nana* relict stands in Poland (Ercan et al., 2021) and is

applied here to a subfossil leaf record from the sediment core of the continental (Eastern Baltic) region for the first time. Compared to the already available alternative temperature proxies for the same sites, this approach uniquely enables the first detailed analysis of seasonality changes during the Lateglacial in the eastern Baltic region.



## 2 Study area

The topography of the eastern Baltic region has been largely shaped by Pleistocene glaciations, where Weichselian glaciation (SIS) in particular contributed to present day topography. The region is currently situated in the hemiboreal vegetation zone, within the transitional zone between the boreal and nemoral forest zones of Europe. Estonia and Latvia are located between 56°N and 59.5°N on the eastern coast of the Baltic Sea in a transitional area from maritime to continental climate, characterized by a west–east gradient in the continentality of the climate. Climatic conditions are mainly determined by high cyclonic activity

and the prevalence of westerlies (Jaagus et al., 2010). The mean annual temperature in Estonia varies between 4.1-6.5 °C (Riigi Ilmateenistus 2021). The mean annual temperature in Latvia is 7.2 °C, ranging from –1.6 °C (mean temperature in February) to 17.7 °C in July (Krauklis and Draveniece, 2004). The snow cover in Estonia persists from December to late March. The mean value of snow cover duration varies from 75 days in the westernmost islands to more than 130 days in the higher and more forested regions in the north-east and south-east (Jaagus, 1997). A continuous snow cover in Latvia forms between

December 8th and 29th, except for the coastal territories in western Latvia where it is established during the first decade of January (Draveniece, 2009) and melts in March/April (Krauklis and Draveniece, 2004).

The vegetation types during the Lateglacial period in the eastern Baltics spanned from arctic tundra in N. Estonia to mixed forests in Latvia (Amon et al., 2014; Veski et al., 2012). The vegetation dynamics is influenced by Lateglacial hemispheric climate fluctuations (Rasmussen et al., 2014) as well as by the local factors (Amon et al. 2014). In the Lateglacial period

(14500–11700 cal y BP) an important palaeogeographical feature of the deglaciated eastern Baltic region was the formation of ice lakes (Vassiljev and Saarse, 2013; Rosentau et al., 2009; Amon et al., 2014). The vegetation communities during the Lateglacial period in the study region, as described by plant macrofossil and pollen records, spanned from pioneer snow-patch tundra to mixed boreal forests (Amon et al. 2012, Amon et al., 2014; Amon et al., 2016).

The present study explores the rich subfossil dwarf birch (*Betula nana*) leaf records from two sites, namely Lake Kosilase and

Lake Lielais Svetinu (L. Svetinu) ~200 km southeast of Lake Kosilase (Fig. 1).

Lake Kosilase (58°20.57 N; 25°39. 48 E) is a small (1.8 ha) lake located in central Estonia. Lake L. Svetinu (surface 18.8 ha) is located in eastern Latvia (56°45.5 N; 27°08.8 E). The postglacial sediments of Lake L. Svetinu have been studied using multiple palaeoecological proxies: chronology and vegetation proxies (Veski et al., 2012), phytoplankton (Stivrins et al., 2015), palaeotemperature reconstructions (Veski et al., 2015), and ancient DNA (Kisand et al., 2018).

## 3 Method


The subfossil leaf material was extracted from the sediments during the plant macrofossil analyses. The sediments were wet-sieved and the remaining material was identified under the stereo- and light microscope. The subfossil *B. nana* leaves were stored in distilled water at 4 °C until a further analysis with the fluorescence microscope. Leaves with preserved cuticle cells were photographed under the microscope Leica using fluorescent light at 200X magnification (Fig. 2). Epidermal cell

properties were measured on digital images using the software ImageJ. From the measured cell area (CA, μm$^2$) and cell





circumference (CC, μm), the undulation index (UI, dimensionless) was calculated according to Kürschner (1997). The UI was determined for a minimum of three epidermal pavement cells on at least three leaf fragments per sample.

From the UI data, growing degree days above the threshold temperature 5 °C (GDD5) and budburst dates were calculated with the GDD5 and budburst inference models for *B. nana*, $GDD_5 = 10^{(2.4232+6.0284*log(UIfossil))}$ and Budburst Day of Year $= 10^{(2.3109-(-1.6888*(log(UIfossil)))}$ respectively, following the procedure outlined in detail in Steinthorsdottir and Wagner-Cremer (2019). Budburst dates are commonly used to determine spring onset. Spring onset dates are subsequently expressed as Day-of-Year (DoY) before Mid-summer Warmth (MSW), referring to the number of days before the 15th of July (DoY: 196), which is set as the date for maximum summer warmth in order to facilitate a comparison with summer temperature proxies such as the chironomid-based July temperature reconstructions (Steinthorsdottir and Wagner-Cremer, 2019). This procedure enables the quantification of the time period between spring onset and maximum summer temperatures by calculating the DoY data rather than providing numbers for individual months.

Two samples from Lake Kosilase were selected for $^{14}$C dates. The selected samples were from the both ends of the ~30-cm-long silty sediment section. The chironomid-based temperature reconstruction from Lake Nakri was published previously (Heiri et al., 2014), as were the pollen-based temperature reconstructions (Veski et al., 2015) and the plant macrofossil data (Veski et al., 2012) from Lake L. Svetinu.

## 4 Results

Radiocarbon dating results for Lake Kosilase are given in Table 1. The radiocarbon dates were calibrated using Oxcal 4.2.4 (Bronk Ramsey, 2009) and the IntCal13 calibration curve (Reimer et al., 2013).

Table 1. $^{14}$C dates from Lake Kosilase, Estonia.

| Core depth, cm | Laboratory code | Dated material | $^{14}$C date, yr BP | Calibrated, cal yr BP | Median calibrated value, cal yr BP |
|---|---|---|---|---|---|
| 966-967 | Poz-104880 | *Dryas octopetala* leaves | 11350 ± 70 | 13070 – 13324 | 13200 |
| 986-987 | Poz-104784 | Wood | 12010 ± 70 | 13720 - 14080 | 13870 |

The mean UI from Lake L. Svetinu subfossil leaf record varies between 1.17 and 1.25 around a total average of 1.2 with an average standard error of 0.03. The UI values of Lake Kosilase subfossil dwarf birch leaves range from mean 1.17 to 1.23 with a total average of 1.21 and a standard error of 0.04 (Fig. 2). The estimated GDD5 range from 674 to 1009 (Fig. 3). The spring onset calculations were based on inferred budburst dates recorded by the fossil leaves and revealed spring onset dates between the 21st of May and June 6th, equivalent to DoY 146 – 157 and translating to 39 – 55 days before MSW (Fig. 3).



## 5 Discussion

The combination of (palaeo)botanical and phenological records reveals the dynamics of the vegetation, thermal growing season duration, and spring onset during the Lateglacial period in the study area. The captured spring onset signal is probably regional. We have a temporal overlap of the two spring onset records from the study sites ~200 km apart. The sample density in the overlapping part is not high, but the general trend (earlier spring onset around 13900 cal yr BP, later spring onset around 13800 cal yr BP, and earlier again around 13600 cal yr BP) observed in both datasets suggests comparable patterns of the spring onset signal over the entire study region.

In the present study, we present spring onset and GDD5 records based on the UI analysis of *B. nana* leaves from two study localities ~200 km apart (Fig. 1). Both lakes are currently surrounded by hemiboreal vegetation, but during the Lateglacial period the floral situation was much more dynamic. During that time Lake L. Svetinu and Lake Kosilase were located in vegetation zones that oscillated north- and southwards, following changing climatic conditions and species migration. The pioneer vegetation phases at both localities were characterized by treeless, *Betula nana* and *Dryas octopetala* dominated tundra. During the Allerød, warming permitted the migration of various tree species to eastern Latvia with traces of mixed forests of birch, pine, and aspen, as evident from the macroremain records (Veski et al., 2012). However, the proximity of the retreating glacier and areas submerged by the Baltic Ice Lake hindered the northward migration of trees and the regional Lateglacial maximum northern treeline was likely located in southern Estonia (Amon et al., 2014). This implies that even during the warmer phases of the Lateglacial, the surroundings of Lake Kosilase most likely remained treeless and were mainly covered by tundra vegetation. During the Younger Dryas cooling the tree remains disappeared from the records of Lake L. Svetinu, suggesting the re-establishment of tundra vegetation and a southward regression of the regional treeline. Although such vegetation shifts are recognized in many sedimentary records, the driving role of growing season dynamics has so far hardly been taken into account, given the lack of temperature records for the early season. Here we have the unique opportunity to synthesise vegetation dynamics and link it to climatic changes through local macroremain analysis, GDD5 and spring onset reconstructions from *B. nana* leaf remains, and more regional temperature signals for winter and summer deduced from pollen and chironomid data.

The length of the growing season is correlated with the UI of both *B. nana* (Wagner-Cremer et al., 2010) as well as *Betula pubescens* subsp. *czerepanovii* (Ercan et al., 2019). The estimated length of the Lateglacial growing season in the Baltic region varies between ~670 and 1000 cumulative growing degree days (GDD5) (Fig 3). It is comparable with current Northern and mid-Finnish conditions (Ercan et al., 2019). The GDD5 estimations from the microphenological fossil dataset in Germany (Schleinsee) stop at 14400 cal yr BP, when the present dataset from the Baltic region starts (Steinthorsdottir and Wagner-Cremer, 2019). The GDD5 estimations from Hässeldala (Sweden) (Steinthorsdottir and Wagner-Cremer, 2019) start at 12500 cal yr BP and span until the start of the Holocene; they are in general lower than the Baltic GDD5 estimations, however both





display an interval of higher GDD5 values during the Younger Dryas cooling period. The Holocene onset is more pronounced in the Hässeldala (Swedish) dataset than in the Baltic (Latvian) one.

The shifting of vegetation zones and migration of the species is evident from the studied *B. nana* record - no *B. nana* leaf remains are recognized in the L. Svetinu core between 13500 - 12500 cal yr BP. Despite the absence of *B. nana* leaves during

this period, the continuous plant macrofossil records document the persisting local flora and provide evidence for vegetation dynamics. The dominant plant species substituting the dwarf birch during the Allerød in Lake L. Svetinu were tree birch (*Betula pendula*) and pine (*Pinus sylvestris*) (Fig 3). The occurrence of these two species provides additional information on the spring conditions through their contemporaneous growth requirements. Phenological observations of *B. nana* budburst dates from the Kevo research station in Finnish Lapland suggest that the number of days from spring onset to MSW varies

between 25 and 56 days, with a mean of 42 days. A modern dataset for *B.pendula* from several study points in Finnish Lapland indicates that the number of days from spring onset to MSW ranges from 40 to 62 days; average spring onset date is 51 days before MSW (Pudas et al., 2008). The modern observations of the budburst dates of *Pinus sylvestris* in the Inari region in Finnish Lapland (Salminen & Jalkanen, 2015) suggest the spring onset range of *P. sylvestris* to be from 50 to 75 days before MSW. Comparison of the modern phenological characteristics to the plant macroremain data thus indicates that the major shift

in vegetation composition was related to spring onset dynamics, which might have taken place earlier in the warmer periods of the Lateglacial.

The notable shift towards earlier spring onset ~13350-13400 cal yr BP (at the end of GI-1c or Allerød) evident from Lake Kosilase coincides with the findings of pine macrofossils in the sediments of Lake L. Svetinu (Fig 3). It suggests an ameliorating, warmer environment in this region. The microphenology of dwarf birch growing in the tundra/at the verge of the

treeline (Kosilase) parallels the warming signal that supported the formation of mixed boreal forests in eastern Latvia, ~200 km southwards (L. Svetinu).

The regional spring warming ~13350-13400 cal yr BP and its effect on past vegetation may be similar to the modern tundra greening process and earlier spring dates reported by modern phenological observations. The warmer climate, including spring, supported the northward shift of the treeline: pine was present in Latvia; ~13300-13000 cal yr BP the scarce tree-birch

macrofossils were found in the sediments of Lake Nakri (~100 km southeast of Lake Kosilase, Amon et al., 2012). The plant macrofossil records suggest that the Lateglacial treeline did probably not reach beyond central Estonia (Amon et al., 2014; Amon et al., 2016), although a single poorly preserved, probably tree-birch or hybrid, macrofossil was found in the Lake Kosilase sediments. At the same time, spring in central Estonia was already taking place much earlier as confirmed by the UI of the *B. nana* leaves and could have been suitable for tree species growth. Thus, the limiting factor for the advancement of

the treeline may have been related to other local conditions in the partially-submerged mosaic landscape, the proximity of the retreating glacier, and the maximum summer temperature. The modern distribution of the treeline-forming tree-birch in Scandinavia follows the 13.2 °C isotherm for summer temperature (Odland, 1996) and the 10–12 °C summer temperature isotherm for tree-line formation (CAVM, 2003). The chironomid-based July temperature estimates around Lake Nakri for this period were ~+12,7-12,8 °C (Heiri et al., 2014) and the pollen-based reconstructed mean May-June-July-August temperature


range was ~+12,2-12,3 °C, which are close to the modern tree-line limits. Assuming that the summer temperature was one of the limiting factors for the northwards migration of the treeline, the results resemble the vegetation response under modern climate conditions. The spring onset thus occurs before a significant rise in the summer temperature. The observed warmer climatic conditions were subsequently disrupted by the onset of the Younger Dryas cold reversal. A similar set – warming of the spring preceding the warming of the summer temperatures – is observed by Steinthorsdottir and Wagner-Cremer, 2019.

**Lateglacial climate fluctuations and seasonality signal**

Pollen-based temperature estimations for May-June-July-August (MJJA) and December-January-February (DJF) are available for Lake L. Svetinu in the Lateglacial period (Veski et al., 2015) and provide an indication of the winter and summer conditions prevailing in the studied region. A minor offset between the pollen record and the microphenological dataset is likely related to the different sampling strategies. The pollen samples were taken as 1-cm-thick slices after some interval while the plant

macrofossil samples from where the dwarf birch leaves were extracted were 5-cm-thick continuous samples. Additional chironomid-based July temperature estimates from Lake Nakri (Fig. 3) ~90 km north are included to complement the temperature data (Heiri et al., 2014).

In the oldest part of the record between 14500-~14000 cal yr BP, assigned to GI-1e (Bølling), all proxies (pollen-based DJF and MJJA temperatures, GDD5, and spring onset estimates (Fig. 3)) follow a comparable pattern of moderate temperatures

during all seasons. A short pulse of cold conditions is paced between 13800 and 13950 cal yr BP with only +10°C for MJJA/chironomid July, dropping to -24°C for DJF; GDD5 values and spring onset (Fig. 3) are declining in parallel. Since this cool pulse is of short duration, the temporal resolution of our proxies is not suitable to determine the offsets in the seasonality changes. The common cooling is interpreted to reflect the GI-1d (Older Dryas) phase (Rasmussen et al., 2014). A gradual decrease in spring thermal conditions has been observed earlier in a GDD5 / spring onset reconstruction from the Schleinsee

section in Southern Germany, where the early season cooling during the Bølling culminates in the Older Dryas, after which the conditions improve again towards the Allerød (Wagner-Cremer and Lotter, 2010). Although the Schleinsee site is located much further to the south, the comparable records point towards a dynamic seasonality pattern during this period.

The following warm period (Allerød) is characterized by a temperature increase occurring in all proxies. The summer-related proxies reveal peaks at ~13700 cal yr BP with +12 °C pollen MJJA and +13.7 °C chironomid J. This initial peak is followed

by a small decrease in the pollen based MJJA and GDD5 / spring onset data which is not picked up by the chironomid record. During the Allerød, *B. nana* disappears from the record while *B. pendula* and *P. sylvestris* become common in the macroremain record. This succession provides additional evidence for stable, mild conditions.

During the Younger Dryas, the signals captured by different proxies diverge. At ~12700 cal yr BP, a rapid temperature decline from ~13 to ~10 °C started according to chironomid-based reconstructed estimates although there is a notable variability

between the individual samples (Fig 3). At the same time (~12650 cal yr BP) the winter (DJF) pollen-based temperature estimates start to decrease as well, reaching the minimal temperature at ~12000 cal yr BP (-20 °C). The minimal chironomid-based temperature estimate (+8.6 °C for July T) is reached ~12500 cal yr BP. This cooling is not entirely paralleled by pollen-



based summer reconstructions and GDD5 / spring onset data: The MJJA pollen-based temperature remains at the previous (Allerød) level, even indicating a small rise between 12500-12100 cal yr BP, only decreasing from 12100 cal yr BP until the

onset of the Holocene. Spring season conditions rather follow the declining winter temperatures than the summer signal. The GDD5 values decrease from the ~12400 cal yr BP peak values of above 1000 GDD5 to low ~720 GDD5 at 12250 cal yr BP. Unfortunately, we do not have good data coverage for this time interval and, thus, cannot provide a detailed analysis on the validity of the peak values nor the exact timing of the cooling. From 12200 cal yr BP the GDD5 values stabilise on very low values, varying between 700–800 GDD5. The results also indicate very late spring onset during this phase, where the growing

season did not start earlier than 39–43 days before MSW (in June). The general pattern during the Younger Dryas in terms of seasonality changes very closely resembles the results obtained for the Swedish Hässeldala section, where GDD5 values ranging from 650–700 were reconstructed, and spring onset occurred ~40 days before MSW (Steinthordottir and Wagner-Cremer, 2019).

The strong variability in several proxies during the Younger Dryas cold reversal (Fig 3) points to a notably dynamic

temperature system. There is an ongoing debate about warm summers on the northern latitudes during the Younger Dryas cooling (Björck et al., 2002; Schenck et al., 2018). The pronounced variability on chironomid-based dataset, duration of the growing season and spring onset data, and rather stable pollen-based MJJA temperatures may indicate short, occasional phases with possible early springs and warmer summer temperatures. However, it does not substantially confirm persistent warm summer conditions during the Younger Dryas as the main trend in chironomids, pollen-based DJF temperatures, and spring

onset / GDD5 records indicate a cooling period. Global climate simulations for the Bølling, Allerød, and Younger Dryas (Schenk et al., 2018) show that for the Baltic region the month of May is highly important, with colder May temperatures during the Younger Dryas than during the Allerød period. That is in broad accordance with the trend in the spring onset record but punctuated by occasional short phases with earlier budburst in the first part of the Younger Dryas. Modern studies suggest that the ongoing lengthening of the vegetation period is also due to the earlier budburst and spring onset (Jeong et al., 2011).

The relevance of the early season temperatures requires proxies that are able to distinguish between the summer/winter temperatures and spring temperatures.

**Conclusions**

- The present study is one of the rare examples of the application of the micro-phenological proxy to the Lateglacial fossil
leaf record. By applying a novel microphenological proxy and plant macrofossil analysis to the multi-proxy approach we can constrain early season temperature dynamics rather than traditionally inferred winter or summer temperatures.
- The combination of spring onset/GDD5 estimates and plant macrofossil records indicates that the local tree-line dynamics are linked to local rather than regional environmental conditions. The spring onset in central Estonia during the Allerød period took place early and was comparable with the timing in the modern forest species. However, the Lateglacial plant
macrofossil data of North Estonia displays the lack of any tree remains. Therefore, we could reconstruct a similar situation



with modern climate change when the spring onset takes place earlier, but the summer temperatures are not yet rising. The explanation for the eastern Baltic Lateglacial situation could be the proximity of the SIS and the Baltic Ice Lake.

- The Lateglacial seasonality dynamics are evaluated by comparing spring onset estimates and pollen-based DJF and MJJA
temperature reconstructions in combination with chironomid-based July temperatures. During the GI-1d and the subsequent warming period (Bølling/Allerød) all proxies agree, while the divergence during the YD suggests a dynamic seasonality pattern with possible occasional warm episodes.

**Author contribution:  LA is the main author (conceptualization, investigation, writing). The co-authors contributed: FW (conceptualization, methodology, writing), JV (chronology), SV (past vegetation and climate reconstructions). LA**
**prepared the manuscript with contributions from all co-authors.**

**The authors declare that they have no conflict of interest.**

**Acknowledgements: This study was supported by Estonian Research Agency grants MOBTP140 and PRG323.**

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

**Figure 1: A. The location of the study sites B. The position of the Scandinavian Ice Sheet, the shoreline of the Baltic Ice Lake and the vegetation zones characterised by dominant plant macroremains ~13300 cal yr BP. The macrofossil data and chronologies for Lake L. Svetinu were published in Veski et al., 2012; for Kosilase the dates are presented in the present paper, plant macrofossil data is unpublished. The chronologies and plant macrofossil data for the other study sites on the**

**map: 1) Nakri (Amon et al., 2012), 2) Prossa (Amon et al., 2014), 3) Kursi (Amon et al., 2016), 4) Udriku (Amon and Saarse 2010). The shoreline reconstruction of the Baltic Ice Lake is modified from Vassiljev and Saarse 2013.**

**Figure 2: The results of the undulation index (UI) measurements. The sample cloud represents the measurements, and the dots display the average values *per* sample. Two sample photos of a *Betula nana* subfossil leaf and cuticle.**





**Figure 3: The compilation of Lateglacial vegetation and climate proxies from the study region: selected plant macrofossil records of Lakes L. Svetinu (L.SV) (Veski et al., 2012) and Kosilase (KOS); pollen-based MJJA & DJF temperature estimations of Lake L. Svetinu (Veski et al. 2015), chironomid-based July temperature estimation of Lake Nakri (location No 1 on Fig3) (Heiri et al., 2014), the spring onset and growth-degree-days dynamics.**



Figure 1.

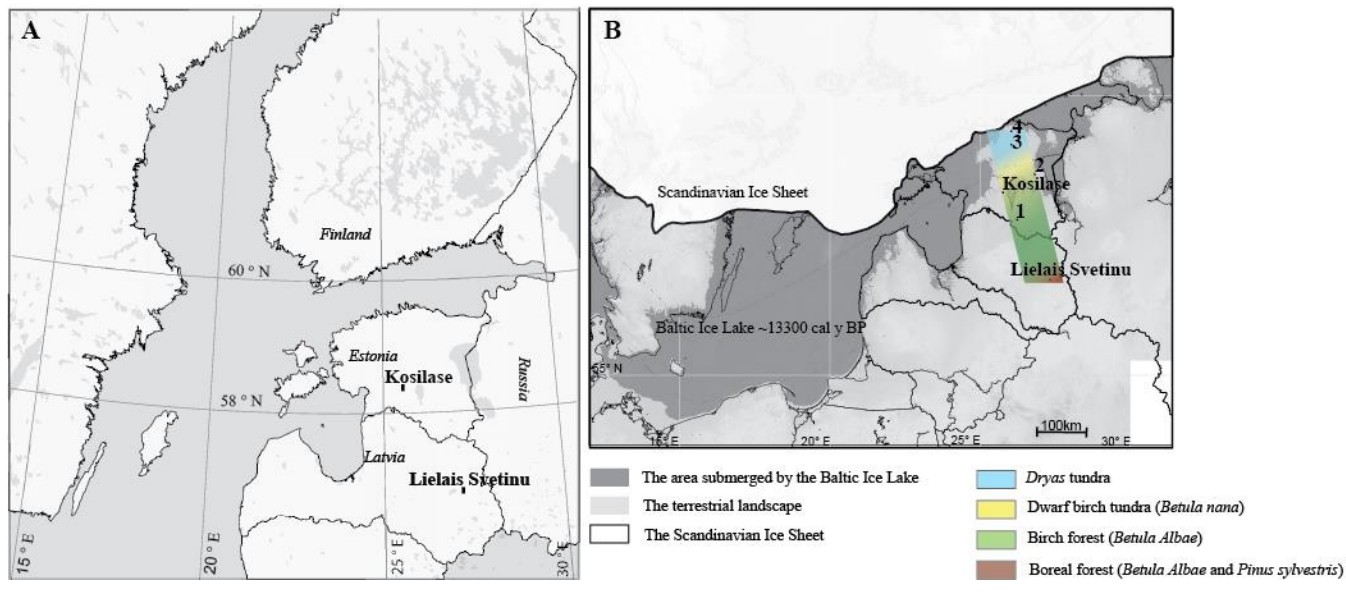



Figure 2.

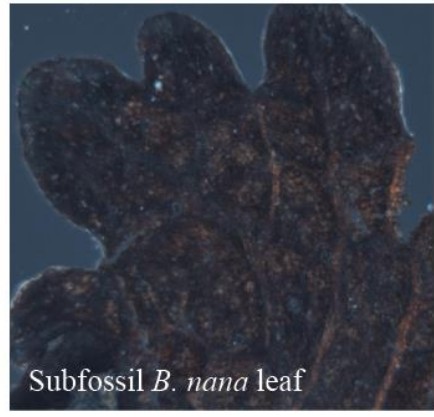

Subfossil *B. nana* leaf

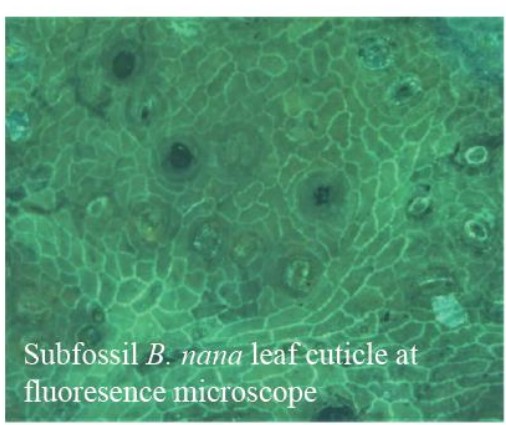

Subfossil *B. nana* leaf cuticle at fluoresence microscope

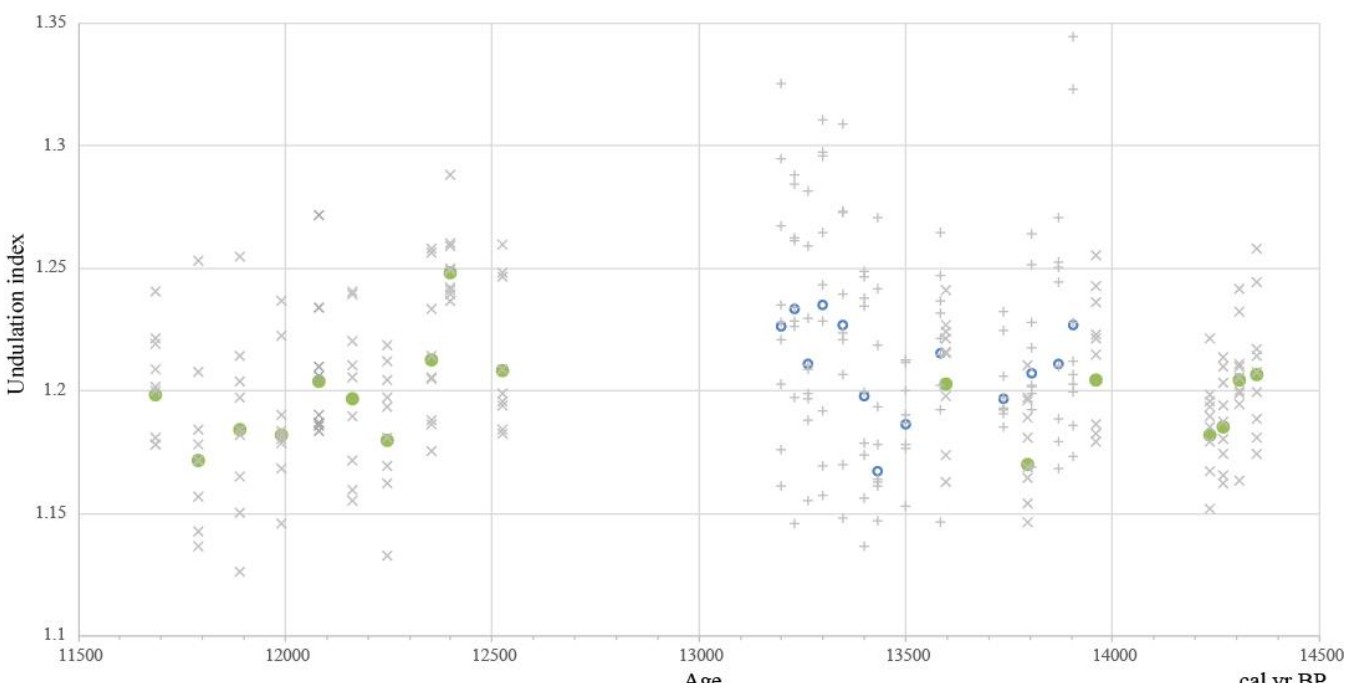

× Measurement point of UI from a fossil *B.nana* leaf from Lake L.Svetinu sediment
+ Measurement point of UI from a fossil *B.nana* leaf from Lake Kosilase sediment
● Average UI from a fossil *B.nana* leaf from Lake L.Svetinu sediment
○ Average UI from a fossil *B.nana* leaf from Lake Kosilase sediment






Figure 3.

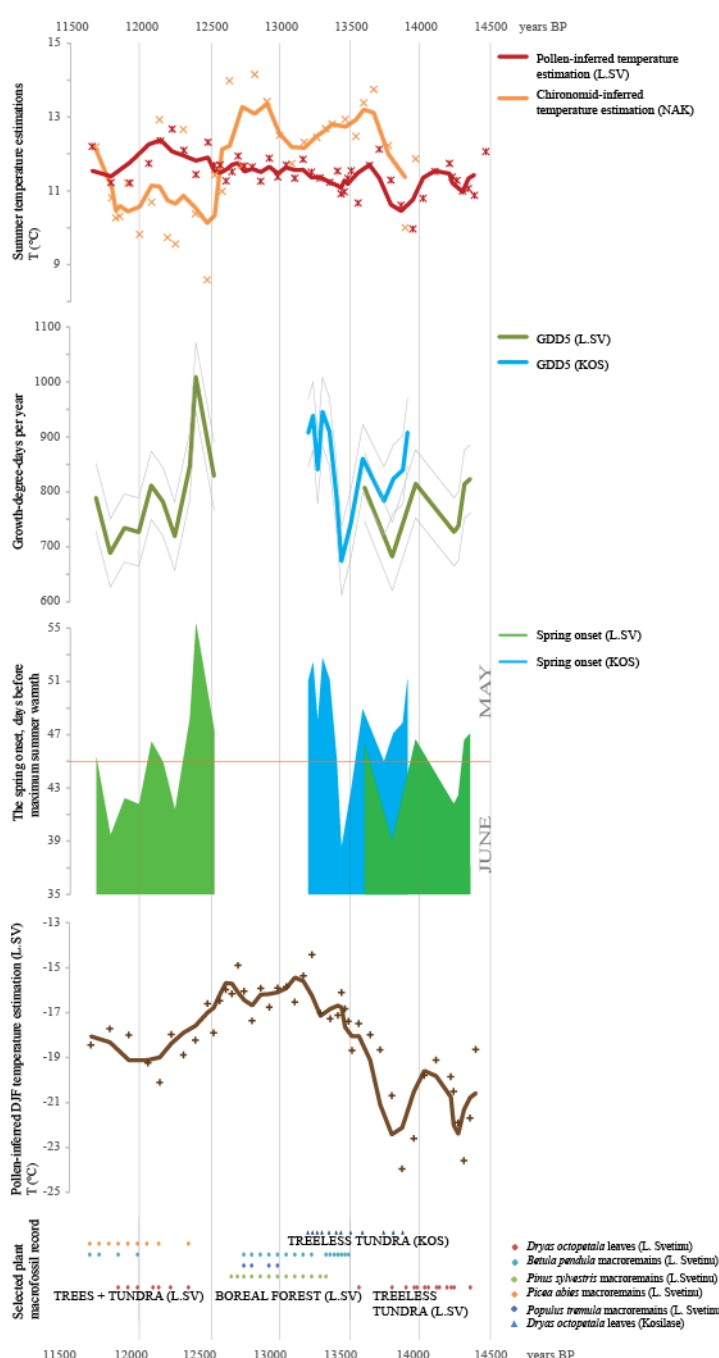