# Peer review of "Spring onset and seasonality patterns during the Lateglacial in the eastern Baltic region"

_Climate of the Past, 2021_

## Author Response (AR1)

Review 2:

We thank Dr Kurina for the support, careful reading, and helpful comments.

Q: Lines 70-75 Description of the study area – I suggest to show modern climate conditions for the two coring lakes in a uniform way.

A: The modern climate data around the two study sites (Estonia: Lake Kosilase and Latvia: Lake Lielais Svetinu) are now presented in more uniform way. We discussed the days of snow cover and decided that we would go with months only: as You noted the variability between the modern observed years may be high. In case of present paleodataset from sediments we are working with much lower resolution, averaged over years.

Q: Line 87-89 – you described what proxies were applied to the sediments from the L. Svetinu Lake. Please, add the information with what proxies the Lake Kosilase was investigated –uniformly as you did for the first one.

A: Lake Kosilase is under study currently – only an unpublished plant macrofossil dataset and two radiocarbon dates, published in the present article are available.

Q: Line 114 – you used IntCal13 curve for calibration. Why did not you use IntCal20 curve, which is newer?

A: We recalibrated and updated all chronologies in the article using IntCal20.

Q: Line 137 – you wrote "During the Allerød, warming permitted the migration of various tree species to eastern Latvia…". Please, clarify from where tree species migrated to Latvia.

A: The studies of Lateglacial treeline advancements in the region (Amon et al. 2014) suggest that the trees came from the southern, south-eastern regions (e.g. Lithuania and Belarus (in case of spruce) where more favorable conditions supported the trees and formation of forests before the environmental conditions allowed the migration of trees to Latvia.

Q: Lines 177-178 – you wrote "similar to the modern tundra greening process and earlier spring dates reported by modern phenological observations". Could you add reference to these modern phenological observations? Next, in the Introduction section (Lines 26-28) you mentioned "The earlier unfolding of leaves has been observed since the mid-20th century all over the Northern Hemisphere (links), although the rate of change has decelerated during the last few decades (link)". I consider, that this phrase (Lines 26-28) contradicts to the phrase from Lines 177-178. Could you explain why the rate of leaf unfolding has decelerated during the last decades? And, next, how does this fact compare to "earlier spring dates", which you wrote in Lines 177-178?

A: The hypothesis of a slowdown in responsiveness raised by Fu et al (2015) has in the meanwhile been shown to be an artifact of analysis strategy (Wolkovich et al., Global Change Biology, 2021;27:4947-4949 doi: 10.1111/gcb.15746). We would like to remove "although the rate of change has decelerated during the last few decades (Fu et al., 2015).". We rephrased the sentences, and added the relevant new references.

Q: Lines 137, 142 and further – in the Discussion you mentioned the names of the known phases (Younger Dryas, Allerød) of the Lateglacial period. Please add to them the age range in brackets.

A: The age range is added at the first mention of the climatic phase in the Discussion.

Q: Figures 2 and 3 – it would be easier to read if you also point the boundaries of the known phases (as example, Younger Dryas, Bølling, Allerød) of the Lateglacial period in these figures.

A: The Late-glacial climate episodes are added to the main discussion figure, Figure 3 and explanation to the caption.

Q: Figure 2 – please, could you add scale to the photo of B. nana leaves?

A: The scale is added to the figures.

Q: Caption to the Figure 1 – why did you select the time slice of 13300 cal yr BP to show the vegetation zones?

A: Yes, it was the most diverse local vegetation along the Baltic transect, showing communities from mixed forest to tundra. Short note added to Fig 1 caption.

Review 3:

We thank the reviewer for the comments.

Q: Firstly, although lake Svetinu is published it would be very useful to have a summary figure of the chronology of this record and the chronological resolution of the palaeoecological data from Svetinu. Secondly, and most importantly, lake Kosilase has a lateglacial chronology based on only two radiocarbon dates, one close to the start of the interstadial and one close to the end.

… I would really like to see the production of an age model for Kosilase, based on an IntCal20 calibration and a recalibraton of the age model for Svetinu using IntCal20.

A: We added the Table 1 and Figure 2 regarding the chronologies. We recalibrated all chronologies in the article using IntCal20. The sediment core of Lake Kosilase is short, we added the description to the text (line 104).

Q: The uncertainty on the calculaton of UI and inferred bud birst dates needs to be expressed in a table in the results.

A: The uncertainties are expressed as GDD5 model (r2=0.68, p = <0.001, RMSE = 62 GDD5), DOY model (r2 = 0.7, p = 0.002, RMSE – 3.8 days) and added in the text (lines 99-100).

---

## Author Response (AR2)

To Editor 14.07.2022

Thank You for the careful evaluation of the article!

Here are the corrections:

1. I added information about the migration and vegetation dynamics.
2. I adjusted the character sizes on Figures 1, 3 and 4. I added note to the caption of Figure 3 regarding the scale on the photographs.
3. The sentences 27-29 "although…": I deleted the hypothesis and reference to Fu et al. 2015 that was questioned by Wolkovich et al., Global Change Biology, 2021;27:4947-4949 doi: 10.1111/gcb.1574). I added new references regarding the modern phenological observations to lines 29 and 188-189. Otherwise, I did not rephrase the sentence.